# *ZmLBD5* Increases Drought Sensitivity by Suppressing ROS Accumulation in Arabidopsis

**DOI:** 10.3390/plants11101382

**Published:** 2022-05-23

**Authors:** Jing Xiong, Weixiao Zhang, Dan Zheng, Hao Xiong, Xuanjun Feng, Xuemei Zhang, Qingjun Wang, Fengkai Wu, Jie Xu, Yanli Lu

**Affiliations:** 1Maize Research Institute, Sichuan Agricultural University, Wenjiang 611130, China; xiongjingsmile@163.com (J.X.); zhangweixiao136@163.com (W.Z.); zhengdan610065@163.com (D.Z.); xiongphd@icloud.com (H.X.); xuanjunfeng@sicau.edu.cn (X.F.); 18283581522@163.com (X.Z.); wdqdjm@126.com (Q.W.); wfk0909@163.com (F.W.); jiexu28@gmail.com (J.X.); 2State Key Laboratory of Crop Gene Exploration and Utilization in Southwest China, Wenjiang 611130, China

**Keywords:** LBD, drought stress, ROS, stomata, maize

## Abstract

Drought stress is known to significantly limit crop growth and productivity. Lateral organ boundary domain (LBD) transcription factors—particularly class-I members—play essential roles in plant development and biotic stress. However, little information is available on class-II *LBD* genes related to abiotic stress in maize. Here, we cloned a maize class-II LBD transcription factor, *ZmLBD5,* and identified its function in drought stress. Transient expression, transactivation, and dimerization assays demonstrated that *ZmLBD5* was localized in the nucleus, without transactivation, and could form a homodimer or heterodimer. Promoter analysis demonstrated that multiple drought-stress-related and ABA response cis-acting elements are present in the promoter region of *ZmLBD5*. Overexpression of *ZmLBD5* in Arabidopsis promotes plant growth under normal conditions, and suppresses drought tolerance under drought conditions. Furthermore, the overexpression of *ZmLBD5* increased the water loss rate, stomatal number, and stomatal apertures. DAB and NBT staining demonstrated that the reactive oxygen species (ROS) decreased in *ZmLBD5*-overexpressed Arabidopsis. A physiological index assay also revealed that SOD and POD activities in *ZmLBD5*-overexpressed Arabidopsis were higher than those in wild-type Arabidopsis. These results revealed the role of *ZmLBD5* in drought stress by regulating ROS levels.

## 1. Introduction

Drought tolerance is a complex trait that involves a series of adaptive changes in the molecular, cellular, physiological, and morphological levels [1]. Reactive oxygen species (ROS) accumulated under drought stress are versatile in plant development and environmental stress responses [2,3]. ROS act as secondary messengers in stress signaling pathways by triggering defensive/adaptive responses to stress at low-to-moderate concentrations, such as stomatal closure, deposition of lignin and cellulose, and modulation of protein activity and gene expression [3,4]. In addition, ABA stimulates the production of H_2_O_2_ through NADPH oxidase in guard cells [4]. In rice, *Abscisic acid, Stress and Ripening5 (ASR5)* is known to potentiate ABA biosynthesis, the expression of peroxidase 24 precursor, H_2_O_2_ accumulation, and stomatal closure, as well as the osmotic and drought tolerance of the seedlings [5]. However, ROS causes growth retardation and eventual cell death once a threshold of ROS concentration is reached [2]. Therefore, ROS detoxification is essential for cell survival, metabolism, and development. Recent studies reveal that increased expression of ROS-scavenging-related genes can improve plants’ drought tolerance. Yang et al. found that drought-tolerant maize seedlings have higher antioxidant activities and, consequently, accumulate fewer ROS than sensitive genotypes when exposed to water deficit [6]. Overexpression of *OsLG3* significantly improves rice’s tolerance to drought stress by triggering the ROS scavenging system, whereas suppression of *OsLG3* results in decreased ROS scavenging activity and increased drought susceptibility [7].

Lateral organ boundary domain (LBD) proteins, defined by a conserved lateral organ boundary (LOB) domain, belong to the plant-specific transcription factor family [8]. The characteristic LOB domain comprises a C-block containing four cysteine residues (CX2CX6CX3C) required for DNA binding, a Gly-Ala-Ser (GAS) block, and a leucine zipper-like coiled-coil motif (LX6LX3LX6L) responsible for protein dimerization [8,9,10]. The variable C-terminal region of LBD confers transcriptional control of downstream gene expression [10]. Based on the conserved regions, most LBD genes belong to class-I, which is characterized by a complete leucine-zipper-like domain, while all members of class-II have an incomplete or no leucine zipper-like coiled-coil motif [8]. Several LBD proteins form homo-interactions and hetero-interactions, such as LBD10, LBD27, LBD18, and LBD33 in Arabidopsis [11,12], and RTCS (rootless concerning crown and seminal roots), RTCL (RTCS-like), IG1, and RS2 in maize [9,13]. According to whole-genome sequencing, Arabidopsis [8], rice [14], maize [15], barley [16], tomato [17], grape [18], Eucalyptus [19], and potato [20] have been identified as harboring 43, 35, 44, 24, 46, 49, 47, and 43 LBD genes, respectively, of which 7, 5, 7, 5, 6, 7, 8, and 8 are LBD class-II family members, respectively. There were significantly fewer LBD class-II members than LBD class-I members. The expression patterns of LBD genes were diverse, revealing the diversity of LBD function [8,14].

Most studies have revealed the function of class-I members, including organ development, plant regeneration, photomorphogenesis, and environmental cue responses. *AtASL4* was first discovered to regulate leaf development in Arabidopsis [8]. AtASL4 interacts with AtAS1 to bind the cis-element of the *KNAT1* promoter, inhibits the expression of *KNAT1*, and promotes the development of leaf primordia and inflorescence [21]. In maize, *IG1 (indeterminate gametophyte1)* interacts with *RS (AtAS1 homolog)* to regulate the development of female gametophytes and the number of tassel branches [13,22]. At*LBD16*, *OsCRL1,* and *ZmRTCS* are involved in root development downstream of the auxin signal transduction pathway [9,23,24,25]. In trees, *LBD* genes also promote stem thickening by accelerating the cell division activity of vascular cambium cells during secondary growth [26,27,28]. In addition to participating in plant growth and development, *LBD* genes are involved in plants’ responses to external biotic and abiotic stresses [10]. In Arabidopsis, the root-specific LBD gene *AtLBD20* inhibits the defense genes *THI2* (*thionin 2.1*) and *VSP2* (*volatile storage protein 2*) via *COI1 (coronatine-insensitive 1)/MYC2*-mediated jasmonic acid (JA) signaling, thereby preventing the damage of the root-invading fungal pathogen *Fusarium oxysporum* in plants. *AtLBD14* regulates the branching of lateral roots through the ABA signaling pathway [29,30]. *AtLBD15* directly binds to the promoter of *AtABI4* (*ABSCISIC ACID INSENSITIVE4*) to activate its expression, resulting in stomatal closure, reduced water loss rate, and enhanced drought tolerance in *AtLBD15*-overexpressed plants [31]. In rice, *OsLBD12-1* directly interacts with the *OsAGO10* promoter to inhibit its expression, resulting in growth retardation, leaf distortion, anther abnormality, and SAM reduction, and *LBD12-1* has a stronger effect on *AGO10* under salt stress [32].

In contrast, reports about class-II members remain limited. The class-II *LBD* genes characterized thus far are mainly involved in metabolism, such as anthocyanin biosynthesis and nitrogen metabolism [33,34,35]. In this study, the role of *ZmLBD5*—a class-II member—in drought response was investigated. Overexpression of *ZmLBD5* in Arabidopsis caused the drought-sensitive phenotype by suppressing ROS accumulation, increasing the stomatal aperture and water loss. Our results suggest that *ZmLBD5* mediates the response of maize seedlings to drought by regulating H_2_O_2_ homeostasis, and is expected to be used in genetically modified crops.

## 2. Results

### 2.1. ZmLBD5 Was Induced by Osmotic Stress in Maize

In the present study, we cloned the class-II LBD gene *ZmLBD5* from inbred B73 maize. The full-length CDS of *ZmLBD5* is 942 bp, and encodes a polypeptide of 313 amino acid residues with a predicted molecular mass of 33.96 kD and a pI value of 6.61. Sequence alignment revealed that *ZmLBD5* contained a typical DNA-binding domain, CX2CX6CX3C, whereas the GAS block and LX6LX3LX6L coiled-coil motif were incomplete, allowing for a distinction between the class-I and class-II members of the LBD family (Figure 1A).

Given that gene expression levels are regulated by promoters, we examined the *ZmLBD5* promoter region (approximately 2000 bp upstream of the first codon). Several stress-response-related cis-acting elements were present in the *ZmLBD5* promoter, including seven ABREs (ABA-responsive elements), three DREs (dehydration-responsive elements), two LTREs (low-temperature-responsive elements), one MBS (MYB-binding site, involved in drought-inducibility), eight MYBRSs (MYB recognition sites), and other light-response elements (Table 1). Therefore, the response of *ZmLBD5* to drought stress was investigated by RT-qPCR using drought- and ABA- treated maize plants. The results showed that the expression of *ZmLBD5* was strongly induced by ABA and drought stress (Figure 1B,C). These results suggest that *ZmLBD5* plays a prominent role in the response to drought stress in maize.

### 2.2. ZmLBD5 Is Localized in the Nucleus, and Could Form Dimers

Understanding the subcellular localization of gene expression products is important for the functional analysis of genes. To determine the subcellular localization of ZmLBD5, ZmLBD5-GFP was transiently expressed in tobacco leaf cells and maize protoplasts under the control of the cauliflower mosaic virus (CaMV) 35S promoter. The strong green fluorescence signal of GFP was mainly distributed in the nucleus and the cytoplasm, whereas the green fluorescence signal of ZmLBD5-GFP was observed in the nucleus, which completely overlapped with the red fluorescence signal of the nuclear localization signal (Figure 2A,B). In addition, GFP fluorescence was observed in the nuclei of the root cells in ZmLBD5-GFP-overexpressed Arabidopsis (Figure 2C).

Given that the GAS block and LX6LX3LX6L coiled-coil motif of class-I members are essential for protein dimerization, and class-II members are characterized by a lack of or an incomplete domain, the ability of ZmLBD5 to dimerize was tested. The full length of ZmLBD5 and its five truncated peptide fragments (A, B, C, AB, and BC) were tested for the interaction with ZmLBD5 itself and another LBD member, ZmLBD33. Fragments A, B, and C represent the N-terminal C-block (CX2CX6CX3C), the GAS and LX6LX3LX6L coiled-coil motifs, and the C-terminal domain, respectively (Figure 2D). Although it was difficult to determine the essential region for the interaction, homo- and heterodimerization were clearly detected through yeast two-hybrid screening (Figure 2E).

### 2.3. Overexpression of ZmLBD5 Decreased Drought Tolerance in Transgenic Arabidopsis

*ZmLBD5* was overexpressed in Arabidopsis to observe its function. Eleven transgenic lines were generated, and the three homozygous lines with the highest expression levels (OE3, OE10, and OE19) were selected for subsequent experiments (Figure 3A). Transgenic lines and the wild-type seeds were exposed to different concentrations of mannitol (0, 200, 250, and 300 mM). The germination rates of *ZmLBD5*-overexpressed plants were comparable to that of the wild type, and it was significantly delayed along with the increase in mannitol concentration (Figure 3B–E). The cotyledon greening rate of the overexpressed lines was significantly lower than that of the wild type under 250 mM and 300 mM mannitol stresses 3 days after germination (Figure 3F,G).

To further characterize the responses of the wild-type and *ZmLBD5*-overexpressed plants to osmotic stress, 5-day-old seedlings were exposed to different concentrations of mannitol (0, 200, 250, and 300 mM) for 7 days. Unexpectedly, there was no significant difference between the wild-type and the transgenic seedlings, except for line OE19 (Figure 4).

To further understand the role of *ZmLBD5* under drought stress, 7-day-old plants were transplanted into the soil and grown for one month. Then, plants were exposed to drought stress for 10 days. Three days after rewatering, *ZmLBD5*-overexpressed plants displayed a higher survival rate than that of the wild type (Figure 5C,D). Under well-watered conditions, the rosette leaf areas of *ZmLBD5*-overexpressed Arabidopsis were significantly larger than those of the wild type (Figure 5A), and the fresh weight increased in *ZmLBD5*-overexpressed seedlings (Figure 5B). These results indicated that *ZmLBD5* promotes seedling growth under normal conditions, and increases drought sensitivity under drought stress.

### 2.4. ZmLBD5 Increased the Water Loss Rate by Enhancing the Stomatal Density and Aperture

We measured the rate of water loss from detached leaves to investigate why *ZmLBD5*-overexpressed seedlings displayed drought sensitivity. The results showed that detached leaves of *ZmLBD5* transgenic seedlings lost water at a greater rate than that of the wild-type plants after 1 h of dehydration (Figure 6A), indicating that *ZmLBD5*-overexpressed seedlings ran out of soil water more rapidly than the wild-type seedlings, leading to earlier wilting. Given that water evaporated mainly through the stomata, the stomatal number and apertures on abaxial leaves were analyzed. The stomatal number and stomatal aperture of *ZmLBD5*-overexpressed plants were both greater than those of the wild-type plants (Figure 6B–D). Thus, the overexpression of *ZmLBD5* enhanced drought sensitivity by increasing the stomatal number and apertures in Arabidopsis under drought conditions.

### 2.5. Overexpression of ZmLBD5 Improved Antioxidant Enzyme Activity and Blocked ROS Accumulation in Arabidopsis

ROS are important molecular signaling and cytotoxic substances in plants’ response to drought stress. The accumulation of H_2_O_2_ is important for stomatal closure [4,5,36]. Therefore, H_2_O_2_ and superoxide anions were investigated using DAB and NBT staining, and H_2_O_2_ content was quantitatively measured by the potassium iodide method. The accumulation of H_2_O_2_ and superoxide anions in the leaves of *ZmLBD5*-overexpressed seedlings was significantly lower than that of the wild-type seedlings (Figure 7A,B,D). Many antioxidant enzymes—such as POD, SOD, and catalase (CAT)—are associated with ROS levels and the tolerance of plants to abiotic stress [3,37]. SOD activity was higher in *ZmLBD5*-overexpressed seedlings than that of the wild-type seedlings under both normal and drought conditions (Figure 7C). The activities of POD and CAT were not different between the wild-type plants and the transgenic plants, except for OE3 (Figure 7E,F).

### 2.6. ZmLBD5 Negatively Regulates Drought-Related Genes’ Expression in Transgenic Arabidopsis

From the above findings, *ZmLBD5* is a negative regulator of plant in drought tolerance. Therefore, we analyzed the expression of several widely reported drought-related genes *(PP2CA, RD17, RD26, DREB2A, RD29A,*
*and RD29B*) in *ZmLBD5*-overexpressed plants and wild-type plants. Under normal conditions, the expression levels of these genes were similar between the transgenic plants and the wild-type plants, except for *PP2CA* (Figure 8). Under drought conditions, the expression levels of these tested genes were remarkably increased in both the transgenic plants and the wild-type plants. However, the expression levels were lower in the transgenic plants than those in the wild-type plants (Figure 8), indicating that *ZmLBD5* suppressed drought-related genes’ expression under drought stress.

## 3. Discussion

The LBD genes are plant-specific transcription factors. According to their GAS and leucine zipper domains, LBD genes are divided into class-I and class-II members. With the publication of genomic data on different species, the distribution, gene structure, and expression pattern of the LBD gene family involved in plants’ development and stress response have been displayed [14,15,38,39,40]. Most class-I LBD genes are involved in root growth, leaf extension, pollen development, plant regeneration, photomorphogenesis, pathogen response, and secondary cell wall development [10,33,35]. However, studies of class-II members are scarce, and these are involved in anthocyanin synthesis, nitrogen metabolism, root development, auxin response, and GA response [33,34,41,42]. In this study, *ZmLBD5* was reported to negatively regulate drought tolerance by decreasing ROS levels and suppressing stomatal closure.

LBD proteins generally function by forming homodimers or heterodimers with themselves or other proteins [43]. In Arabidopsis, the dimerization of AtLBD16 and AtLBD18 is critical for lateral root formation [44]. AtLBD10 interacts with AtLBD27 to regulate pollen development [11]. In maize, RTCS and RTCL form heterodimers to affect the initial crown root generation [9]. The difference between class-I and class-II LBD proteins is in the GAS and leucine zipper domains, which are proposed to be necessary for protein–protein dimerization [10,35]. In this study, ZmLBD5, without the complete GAS and leucine zipper domains, could form homodimers and heterodimers in the same way as class-I members, implying that the GAS and leucine zipper domains may be not essential for dimerization.

Previous studies have shown that the functions of the class-I LBDs are mainly related to organ determination, including embryo, root, leaf, and inflorescence development [13,45,46,47]. In Arabidopsis, *AtASL4* is expressed at the boundary between the developing leaf primordia and the shoot-tip meristem to regulate leaf development [45]. *AtAS2 (At LBD6)* represses cell proliferation in the adaxial domain, and is critical for the development of properly expanded leaves [46,47]. *IG1*, an LBD gene, affects the formation of the leaf’s ligular region by inhibiting the expression of *KNOX* while also inhibiting the development of female gametophytes in *ig1* mutants, and limiting the number of male ear branches in maize [13]. In addition, the class-I member *AtLBD15* promotes drought tolerance by increasing stomatal closure and reducing the water loss rate [31]. *OsLBD12-1* reduces SAM size by directly binding to the promoter region, and strongly represses the expression of *AGO10* under salt conditions [32]. *ZmLBD5,* a class-II LBD gene, also regulates organ development and response to drought stress in transgenic Arabidopsis, indicating the functional overlap between class-I and class-II members. The function of *ZmLBD5* is similar to that of the class-I members of the LBD gene family, thereby portraying substantial evidence for the study of class-II members in plant growth and the regulation of drought.

Drought stress brings about the production of ROS. The excessive accumulation of ROS causes damage to plant cells; however, they also act as signaling molecules that participate in the regulation of stomatal number and apertures. Therefore, ROS have been widely reported to regulate drought tolerance in various studies [5,6,7,48]. Many studies have shown that ABA stimulates the production of H_2_O_2_ through NADPH oxidase in guard cells, and H_2_O_2_ is an important signaling molecule that activates calcium channels in the plasma membrane, thereby mediating ABA-induced stomatal closure [36,49,50]. In rice, H_2_O_2_ accumulated in guard cells in *dst* (drought- and salt-tolerant transcription factor) mutants was able to promote stomatal closure, reduce water loss, and improve drought and salt tolerance. *DST* also inhibited stomatal closure by directly regulating the expression of the H_2_O_2_ homeostasis-related gene peroxidase 24 predictor [5,48]. In this study, the activities of SOD and POD were remarkably higher, and the ROS level was significantly lower in *ZmLBD5*-overexpressed Arabidopsis than that in the wild type. This indicates that *ZmLBD5* regulates the response of the seedlings to drought through the regulation of H_2_O_2_ signal molecules on the stomatal aperture—not the antioxidant pathway. Accordingly, *ZmLBD5* overexpression increased water loss rate by suppressing stomatal closure, and resulted in the drought-sensitive phenotype.

## 4. Conclusions

In summary, this study demonstrated *ZmLBD5* as a negative regulator of drought tolerance. Overexpression of *ZmLBD5* increased the stomatal aperture and water loss rate by suppressing ROS accumulation. Furthermore, the enhancement of SOD and POD activities in *ZmLBD5*-overexpressed Arabidopsis revealed the role of *ZmLBD5* in drought stress by regulating ROS levels. The study of *ZmLBD5* promoted our understanding of the function of the class-II LBD genes in maize.

## 5. Materials and Methods

### 5.1. Plant Materials and Growth Conditions

*ZmLBD5* transgenic lines and wild-type (ecotype: Col-0) seeds were surface-sterilized with 3% NaClO_3_ for 8 min, and washed five times with sterile water. The sterilized Arabidopsis seeds were plated on half of the Murashige and Skoog (1/2 MS) medium and stored at 4 °C for 72 h in darkness. Then, they were transferred into a growth incubator (22 °C, 16 h light/8 h dark) for germination and growth. Seven days later, the seedlings were transplanted into soil and grown in a greenhouse (22 °C, 16 h light/8 h dark). Tissues were harvested from the seedlings for further study.

For drought stress and ABA treatment, two-leaf-stage seedlings were transferred to Hoagland nutrient solution in a greenhouse with a 14 h light/10 h dark photoperiod at 28 °C, and grown to the three-leaf stage. Then, seedlings were subjected to polyethylene glycol 6000 (PEG6000) (20% *w/v*) and ABA (10 μM). The roots were collected after 0, 1, 3, 6, 12, and 24 h of treatment. The harvested samples were frozen immediately in liquid nitrogen and used for RNA isolation.

### 5.2. Sequence Analysis

The *ZmLBD5* cDNA was obtained from MaizeGDB (https://www.maizegdb.org/) (accessed on 16 February 2022). Homologous sequences of ZmLBD5 were retrieved from the Phytozome database (https://phytozome-next.jgi.doe.gov/) (accessed on 16 February 2022), and sequence alignment was performed using ClustalX. Plant CARE (http://bioinformatics.psb.ugent.be/webtools/plantcare/html/) (accessed on 1 March 2022) was used to analyze the promoter sequences of different abiotic-stress-related cis-elements.

### 5.3. Subcellular Localization

The full-length CDS of *ZmLBD5* was inserted into the binary vector pCAMBIA2300-eGFP to generate the pCAMBIA2300-ZmLBD5-eGFP vector. The constructed vector was introduced into Arabidopsis and tobacco leaves using agrobacterium-mediated methods. In addition, the plasmid was transformed into maize protoplasts via PEG-mediated methods. GFP fluorescence was investigated using a laser confocal microscope (LSM800, Zeiss, Germany). The primers used here are listed in Appendix A.

### 5.4. RNA Extraction and Quantitative RT-qPCR Analysis

Total RNA was extracted from Arabidopsis or maize (inbred line: B73) seedlings according to the manufacturer’s protocol of the Plant Total RNA Isolation Kit (FOREGENE, Re-05014), and treated with DNase I (Trans, GD201-01) at 37 °C for 30 min to eliminate genomic DNA contamination. The PrimeScript RT reagent kit with a gDNA eraser (Takara, RR047A) was used to synthesize first-strand cDNA as real-time PCR templates with gene-specific primers. Furthermore, we performed qPCR amplification on a Bio-Rad CFX96 PCR instrument according to the SYBR Green Fast qPCR Mix Kit instructions (ABclonal, RM21203). *AtACTIN8* and *AtUBQ10* were used as internal reference genes for Arabidopsis. *ZmeF1α* and *Zm18S* were used as internal reference genes for maize. The mean values and standard deviations were estimated using 2^−∆∆CT^ from the data of three biological experiments. All of the primers used in the experiments are listed in Appendix A.

### 5.5. Generation of Transgenic Plants and Phenotypic Analysis

The coding sequence of *ZmLBD5* was cloned into the pCAMBIA2300-eGFP vector to generate the 35S::ZmLBD5-eGFP vector. The product was introduced into the wild-type Col-0 using the agrobacterium-mediated floral-dip method. Homozygous plants were screened under 10 µg/mL neomycin (G418) conditions, and three high-expression lines were used for further study. All of the primers used in the experiments are listed in Appendix A.

Seeds were sterilized and sowed on 1/2 MS medium containing 0 mM, 200 mM, 250 mM, and 300 mM mannitol and grown in a greenhouse. Germination rates were recorded every 12 h. Seeds were recorded as germinated when the radicles protruded from the seed coat. After germination for 5 d, the greening rate of the seedlings was recorded. Each sample contained three biological replicates. Statistical analysis was performed with one-way ANOVA, and the mean value each transgenic line was compared with the wild type.

Five-day-old seedlings vertically grew on 1/2 MS medium containing 0 mM, 200 mM, 250 mM, and 300 mM mannitol for one week. Primary root length was measured using ImageJ software. Root lengths and surface areas were collected and analyzed using an Epson 11,000 × l root scanner and WinRHIZO pro2013. All experiments were performed in triplicate. Each biological replicate contained at least 12 seedlings.

Seven-day-old plants were transplanted into the soil under short-day conditions to grow for three weeks. Subsequently, water was withheld for approximately 10 days, and photographs were taken. After re-watering for 3 days, the survival rates were investigated.

### 5.6. Water Loss Measurement

To assay the water loss rate, 12 one-month-old seedlings of each sample were detached on the laboratory bench and weighed at different time points. The experiment was replicated three times. ANOVA was used to assess the differences between the wild-type and transgenic plants.

### 5.7. Stomatal Density and Stomatal Aperture

The fourth expanded rosette leaves of Arabidopsis were detached on a laboratory bench for one hour. Leaves were placed into the Carnot fixed solution (absolute ethanol: glacial acetic acid = 3:1) for 24 h, followed by dehydration with 30%, 50%, 70%, 80%, 85%, 90%, 95%, and 100% alcohol for 30 min, dehydration with 100% alcohol again, and placement into a transparent solution (trichloroacetaldehyde: water: glycerol = 8:3:1). Stomatal apertures were observed under microscopes, and the ratio of the stomatal length to width was recorded using ImageJ software. At least 30 stomata of each sample per replicate were measured, and three replicates were performed.

### 5.8. ROS Measurements

Histochemical assays for reactive oxygen species (ROS) accumulation were performed using DAB and NBT staining. The detached leaves of Arabidopsis seedlings were treated with DAB staining solution (0.1 g/mL DAB, PH 3.8) through a vacuum pump for 30 min, and placed in the dark at room temperature for 10 h, soaked in decolorizing solution (acetic acid: glycerol: ethanol = 1:1:3) in 95 °C boiling water for 5 min, stored in 95% ethanol, and observed under a stereomicroscope. Superoxide anion accumulation was detected using NBT staining. The detached leaves of Arabidopsis were immediately immersed in 50 mM phosphate buffer (pH 7.5) containing 0.1 g/mL NBT at room temperature for 8 h in the dark. The decolorization method was similar to that of DAB staining. Each line contained at least 10 different seedlings, and representative images are shown.

Quantitative measurement of H_2_O_2_ concentration was performed using the potassium iodide method [51]. Briefly, 100 mg leaf samples were placed in liquid nitrogen and ground into powder. Furthermore, 1 mL of precooled 0.1% trichloroacetic acid (TCA) solution was immediately added and mixed with the samples. After centrifugation (10,000× *g*, 4 °C, 10 min), equal volumes of PBS buffer were added to the 500 µL supernatant, and then 1 mL of 1 M potassium iodide solution was added, and the mixture was shaken with 150 rpm at 30 °C for 1 h. Absorbance was measured at 390 nm. In addition, 300 µmol/L H_2_O_2_ was used to obtain a standard curve. Each experiment was performed in six replicates.

The activities of antioxidant enzymes (SOD, POD, and CAT) were measured following the aforementioned protocols [52,53,54]. The units of the antioxidant enzyme activities were defined as follows: a unit of SOD activity is the quantity of enzyme required to cause 50% inhibition of the photochemical reduction of NBT per minute at 560 nm [54]; a unit of POD activity is the amount of enzyme required to cause a 0.01 increase in the absorbance of H_2_O_2_ per minute at 470 nm [52,53]; and a unit of CAT activity is the amount of enzyme required to cause a 0.01 decrease in the absorbance per minute at 240 nm [53].

## Figures and Tables

**Figure 1 plants-11-01382-f001:**
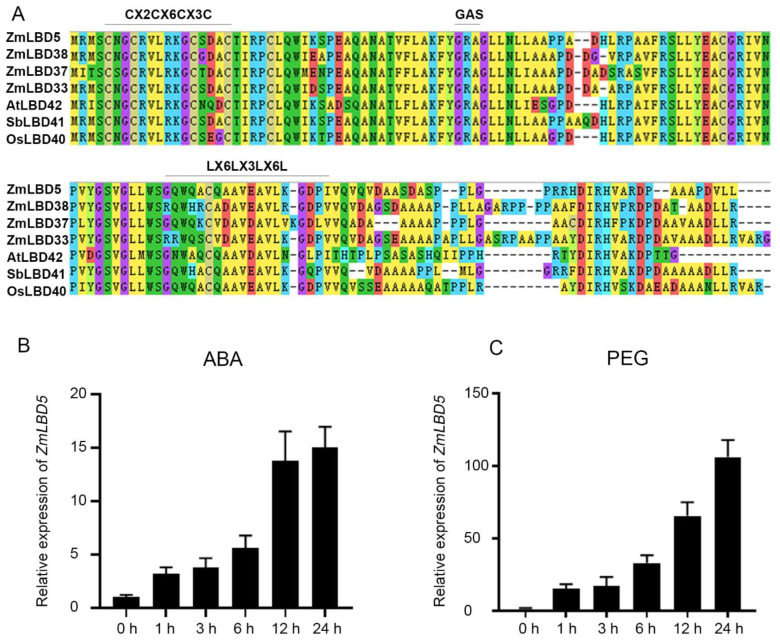
Sequence and expression pattern analysis of *ZmLBD5*: (**A**) LOB domain sequence alignment of ZmLBD5 and LBD members from other plant species. The class-I LBD members had typical CX2CX6CX3C, GAS, and LX6LX3LX6L domains. The same position in class II is marked. (**B**,**C**) The expression of *ZmLBD5* upon ABA and drought treatment in maize. Total RNA was isolated from 3-leaf seedlings grown without (0 h) or with 10 µM ABA and 20% PEG6000 treatment. Transcript levels of *ZmLBD5* were determined by qPCR, using *Zme1F1α* and *Zm18S* as reference genes. Fold change was calculated by 2^−∆∆t^. All bars represent means ± SD (*n* = 3).

**Figure 2 plants-11-01382-f002:**
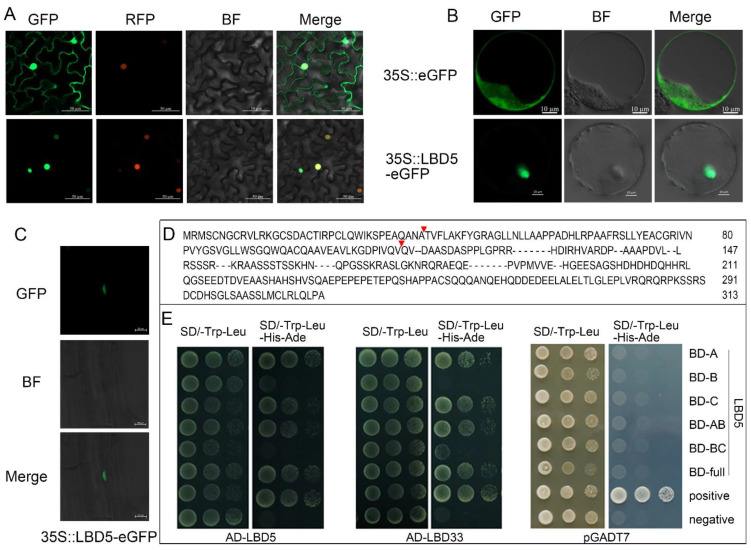
Subcellular localization and dimer-forming ability of ZmLBD5: (**A**–**C**) Subcellular localization of ZmLBD5 in tobacco leaves, maize protoplasts, and 35S::ZmLBD5-eGFP transgenic Arabidopsis, respectively (bar = 50, 10, and 20 μm, respectively). (**D**) Three fragments of ZmLBD5 (**A**–**C**) were divided and the red triangle indicated the position of truncation. (**E**) The ability of ZmLBD5 to form dimers in the yeast strain Y2H Gold. The Y2H Gold strains containing target plasmids were diluted and cultured on no-selection synthetic dropout (SD) media without tryptophan and leucine (SD/-T-L), or on selection SD media without tryptophan, leucine, histidine, and adenine (SD/-T-L-H-A). Photos were taken 3 days after inoculation for the plates. Fragments A, B, and C represent CX2CX6CX3C, GAS, and LX6LX3LX6L, and various C-terminal domains, respectively.

**Figure 3 plants-11-01382-f003:**
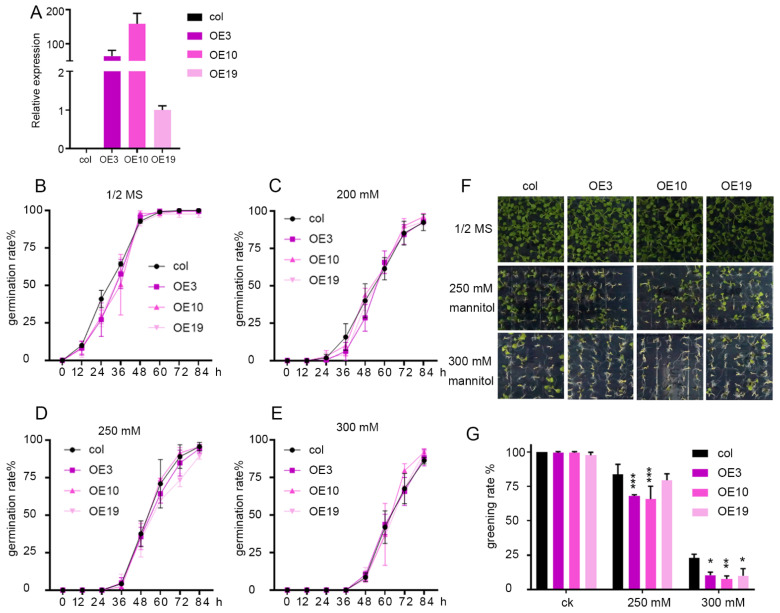
The germination and greening rate of *ZmLBD5* transgenic Arabidopsis with mannitol treatment: (**A**) RT-qPCR analysis of *ZmLBD5* expression in the overexpression lines and wild type. (**B**–**E**) Statistical analysis of *ZmLBD5* transgenic lines’ and wild-type’s seed germination rates under 0 mM, 200 mM, 250 mM, and 300 mM mannitol, respectively. (**F**) The phenotypes of *ZmLBD5* transgenic and wild-type Arabidopsis lines treated with 0 mM, 250 mM, and 300 mM mannitol. (**G**) Greening rate of *ZmLBD5* transgenic and wild-type Arabidopsis under 0 mM, 250 mM, and 300 mM mannitol stress. Significance was calculated by one-way ANOVA. * *p* < 0.05; ** *p* < 0.01; *** *p* < 0.001. All bars represent means ± SD, (*n* ≥ 3).

**Figure 4 plants-11-01382-f004:**
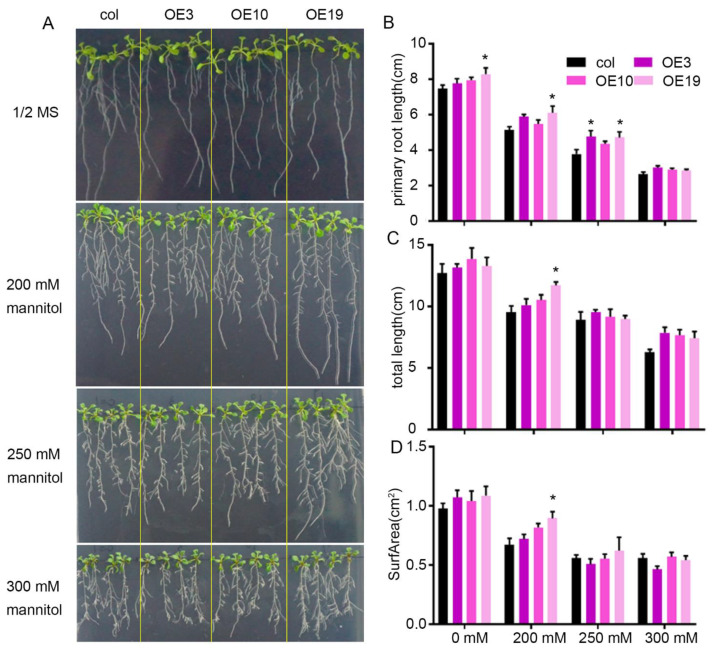
The phenotype of *ZmLBD5* transgenic Arabidopsis with mannitol treatment: (**A**) Seedlings of wild-type and *ZmLBD5* transgenic lines grown on 1/2 MS medium with 0 mM, 200 mM, 250 mM, and 300 mM mannitol. (**B**–**D**) Statistical analysis of primary root length, total root length, and root surface area of wild-type and *ZmLBD5* transgenic seedlings grown on 1/2 MS medium with or without mannitol treatment. Mean values and standard errors (bars) are shown from 12 independent seedlings. Asterisks on bar represent the difference compared with wild type is significant (* *p* < 0.05).

**Figure 5 plants-11-01382-f005:**
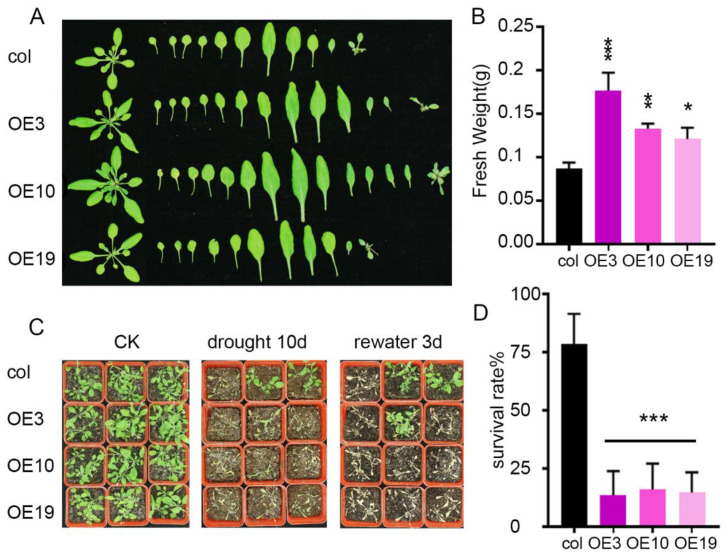
Phenotype and survival rate of *ZmLBD5* transgenic seedlings under normal or drought conditions in soil: (**A**,**B**) Phenotype and fresh weight of *ZmLBD5* transgenic seedlings under normal conditions in soil. (**C**,**D**) Survival rate of *ZmLBD5* transgenic seedlings under drought conditions in soil. Significance was analyzed by one-way ANOVA. * *p* < 0.05; ** *p* < 0.01; *** *p* < 0.001. All bars represent means ± SD (*n* ≥ 12).

**Figure 6 plants-11-01382-f006:**
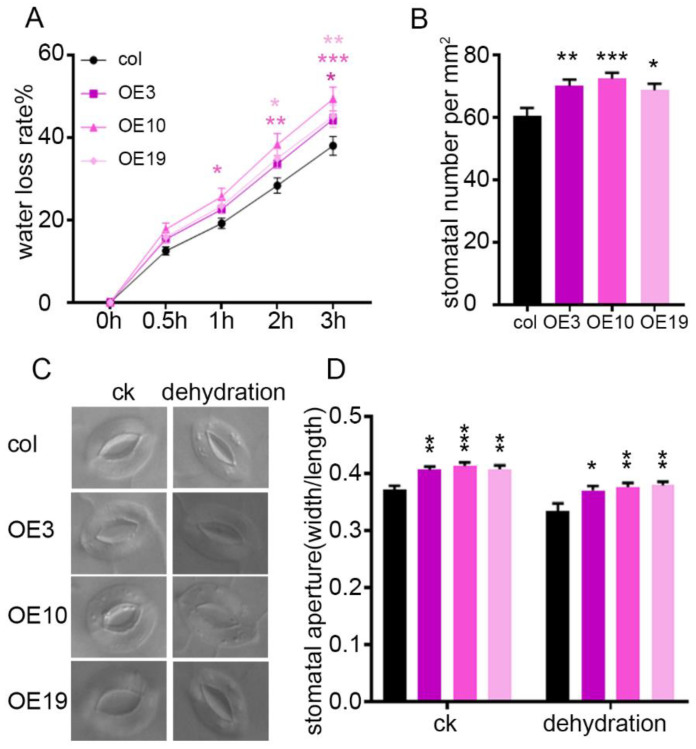
Water loss rate, stomatal number, and apertures of leaves in *ZmLBD5* transgenic seedlings: (**A**) Water loss rate of leaves in *ZmLBD5* transgenic seedlings. (**B**) Stomatal number of the fourth leaves in *ZmLBD5* transgenic seedlings. Each sample had at least 12 seedlings. (**C**,**D**) Stomatal aperture of fourth leaves after detachment for 1 h in *ZmLBD5* transgenic seedlings (*n* > 30 for each sample). Significance was analyzed by one-way ANOVA. * *p* < 0.05; ** *p* < 0.01; *** *p* < 0.001. All bars represent means ± SD (*n* ≥ 12).

**Figure 7 plants-11-01382-f007:**
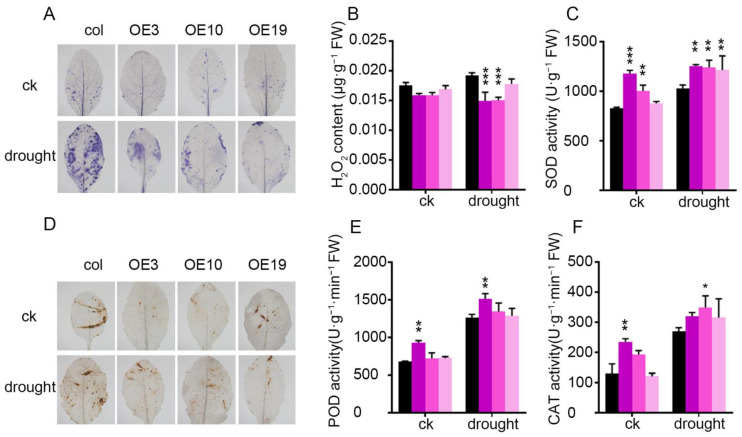
Reactive oxygen species staining and physiological indices in *ZmLBD5* transgenic Arabidopsis: (**A**,**D**) NBT and DAB staining of leaves for H_2_O_2_ from *ZmLBD5* transgenic seedlings and wild-type plants under normal conditions and drought stress (3-week-old seedlings were subjected to drought for 7 days). (**B**) H_2_O_2_ content in leaves from *ZmLBD5* transgenic seedlings and wild-type plants under normal conditions and drought stress (withholding water for 7 days). (**C**) SOD activity, (**E**) POD activity, and (**F**) CAT activity in leaves from *ZmLBD5* transgenic seedlings and wild-type plants under normal and drought conditions (withholding water for 7 days). Significance was analyzed by one-way ANOVA. * *p* < 0.05; ** *p* < 0.01; *** *p* < 0.001. All bars represent means ± SD (*n* = 6).

**Figure 8 plants-11-01382-f008:**
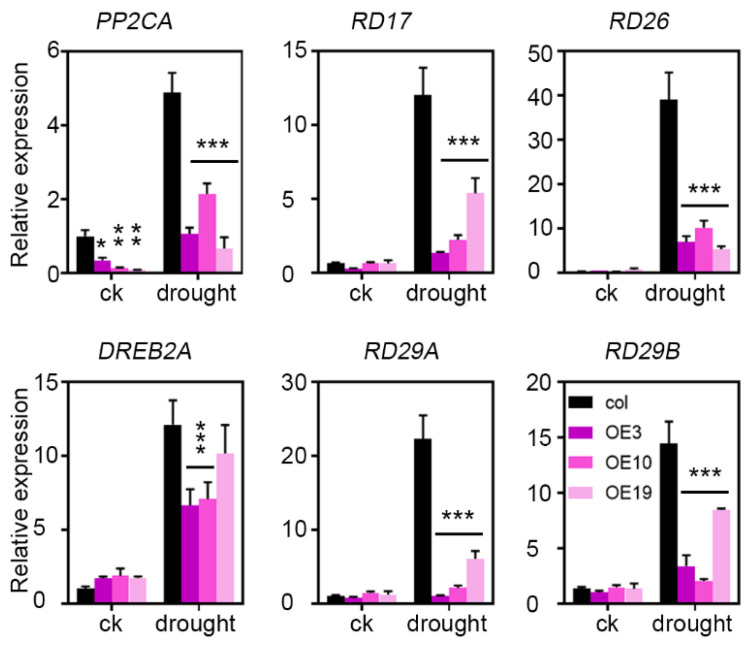
Expression levels of drought-stress-related genes in *ZmLBD5* transgenic Arabidopsis: Total RNA was isolated from 15-day-old seedlings grown without (CK) or with 250 mM mannitol treatment for 7 days. Transcript levels of *PP2CA*, *RD17*, *RD26*, *DREB2A*, *RD29A*, and *RD29B* in the transgenic lines and wild type were determined by qPCR, using *AtACTIN8* and *AtUBQ10* as reference genes. Fold change was calculated by 2^−∆∆t^. Significance was analyzed by one-way ANOVA. * *p* < 0.05; ** *p* < 0.01; *** *p* < 0.001. All bars represent means ± SD (*n* = 3).

**Table 1 plants-11-01382-t001:** Cis-elements in the promoter region (~2 kb) of ZmLBD5.

Site Name	Sequence	Position	Strand	Function
ABRE	ACGTG	−1984	-	Abscisic acid responsiveness
ABRE	CGTACGTGCA	−1730	-	Abscisic acid responsiveness
ABRE	CACGTG	−1596	+	Abscisic acid responsiveness
ABRE	ACGTG	−1595	+	Abscisic acid responsiveness
ABRE	ACGTG	−1528	+	Abscisic acid responsiveness
ABRE	ACGTG	−71	-	Abscisic acid responsiveness
ABRE	CCACGTGG	−1597	+	Abscisic acid responsiveness
DRE	GCCGAC	−1896	-	Dehydration-responsive element
DRE	GCCGAC	−1495	-	Dehydration-responsive element
DRE	ACCGAGA	−38	+	Dehydration-responsive element
LTR	CCGAAA	−1635	+	Low-temperature responsiveness
LTR	CCGAAA	−262	+	Low-temperature responsiveness
MBS	CAACTG	−597	-	MYB-binding site involved in drought-inducibility
MYBRS	CAACCA	−1566	-	MYB recognition site
MYBRS	CAACTG	−597	-	MYB recognition site
MYBRS	TAACCA	−593	-	MYB recognition site
MYBRS	CAACCA	−518	+	MYB recognition site
MYBRS	CAACCA	−100	+	MYB recognition site
MYBRS	CAACCA	−96	+	MYB recognition site
MYBRS	CCGTTG	−1844	+	MYB recognition site
MYBRS	TAACCA	−593	-	MYB recognition site
CCAAT-box	CAACGG	−1844	-	MYBHv1-binding site
ARE	AAACCA	−1631	+	Anaerobic induction
ARE	AAACCA	−259	+	Anaerobic induction
G-box	CACGTC	−1984	+	Light responsiveness
G-box	GCCACGTGGA	−1598	+	Light responsiveness
G-box	CACGTG	−1596	+	Light responsiveness
G-Box	CACGTG	−1596	+	Light responsiveness
G-Box	CACGTT	−1529	-	Light responsiveness
G-box	CACGTC	−71	+	Light responsiveness

“+” represents sense strand, and “-” represents the antisense strand.

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
