# Peer review of "ZmLBD5 Increases Drought Sensitivity by Suppressing ROS Accumulation in Arabidopsis"

_plants, 2022, doi:10.3390/plants11101382_

Round 1
Reviewer 1 Report
In this study, the authors demonstrated ZmLBD5 as a negative regulator of drought stress. They proved it in Arabidopsis, but do they plan to overexpress and functionally characterize ZmLBD5 in maize?
English is fine but there are a few grammatical mistakes, so needs a thorough revision.
Besides, please include a concluding sentence in the end of introduction.
The discussion is very concise. Extend the discussion section by comparing/contrasting your results with those of previous studies. There should be some concluding sentences at the end of discussion section.
I can't see a conclusion section. Please include a separate conclusion at the appropriate site, and clearly state the conclusions from your study.
The manuscript could be reconsidered provided with addressing these concerns.
Author Response
We greatly appreciate the reviewers’ constructive comments and suggestions. We are trying our best to address reviewers’ concerns and improve the manuscript. We fully expect that our response and the presented work can be accepted by the editor and the reviewers. The revised sections are highlighted with the red color in the revised manuscript. Please see our point-by-point response to reviewers’ comments below:
Reviewer 1 Comments for the Author
In this study, the authors demonstrated ZmLBD5 as a negative regulator of drought stress. They proved it in Arabidopsis, but do they plan to overexpress and functionally characterize ZmLBD5 in maize?
Response: Thank for your constructive suggestion. We have characterized the function of ZmLBD5 in maize, and relevant study have been prepared and submitted to Plant Physiology. However, the impact of the COVID-19 pandemic seriously hindered the progress of the review. The presented work in Arabidopsis confirmed the function of ZmLBD5 in maize, and provided a new relationship of ZmLBD5 with ROS, which was not discussed in maize. Therefore, we believe that the results in Arabidopsis can confirm and further supplement the function of ZmLBD5 in different species.
English is fine but there are a few grammatical mistakes, so needs a thorough revision.
Response: We have checked and revised the whole manuscript carefully for the grammatical mistakes, which were tracked with red color.
Besides, please include a concluding sentence in the end of introduction.
Response: We have added a concluding sentence in the end of introduction, which was tracked with red color.
The discussion is very concise. Extend the discussion section by comparing/contrasting your results with those of previous studies. There should be some concluding sentences at the end of discussion section.
Response: We have improved the discussion in the revised manuscript, which was tracked with red color.
I can't see a conclusion section. Please include a separate conclusion at the appropriate site, and clearly state the conclusions from your study.
Response: We thank the reviewer for pointing out the shortcomings of our manuscript. According to reviewer’s suggestion, we have added a separate conclusion in the behind of discussion section, which was marked with red color.
Reviewer 2 Report
This is a well written report of of a new role for the protein products class II LBD transcription factor genes. It provides a rounded account of its role in drought tolerance, fitting with previous knowledge about this class. Measurements of induction of expression, analysis of promotor motifs, sub-cellular localisation, yeast- 2-hybrid analysis and other methods were used, well described and the data is convincing.
Author Response
We thank the reviewers for the positive comment on our work.
Round 2
Reviewer 1 Report
The authors have addressed all my concerns. The modified manuscripts now fulfils the criteria to be considered for publication in this journal.
Author Response
Reviewer 1 Comments for the Author
The authors have addressed all my concerns. The modified manuscripts now fulfils the criteria to be considered for publication in this journal.
Response: We thank the reviewers for their satisfaction with the revised manuscript.
